# Relevance of Sugar Transport across the Cell Membrane

**DOI:** 10.3390/ijms24076085

**Published:** 2023-03-23

**Authors:** Roxana Carbó, Emma Rodríguez

**Affiliations:** 1Cardiovascular Biomedicine Department, Instituto Nacional de Cardiología Ignacio Chávez, Juan Badiano #1, Col. Sección XVI, Tlalpan, Mexico City 14080, Mexico; 2Cardiology Laboratory at Translational Research Unit UNAM-INC, Instituto Nacional de Cardiología Ignacio Chávez, Juan Badiano #1, Col. Sección XVI, Tlalpan, Mexico City 14080, Mexico; emma.rodriguez@cardiologia.org.mx

**Keywords:** sugar transporters, transmembrane communication, transport-derived diseases, therapeutics

## Abstract

Sugar transport through the plasma membrane is one of the most critical events in the cellular transport of nutrients; for example, glucose has a central role in cellular metabolism and homeostasis. The way sugars enter the cell involves complex systems. Diverse protein systems participate in the membrane traffic of the sugars from the extracellular side to the cytoplasmic side. This diversity makes the phenomenon highly regulated and modulated to satisfy the different needs of each cell line. The beautiful thing about this process is how evolutionary processes have diversified a single function: to move glucose into the cell. The deregulation of these entrance systems causes some diseases. Hence, it is necessary to study them and search for a way to correct the alterations and utilize these mechanisms to promote health. This review will highlight the various mechanisms for importing the valuable sugars needed to create cellular homeostasis and survival in all kinds of cells.

## 1. Introduction

Sugar typically refers to simple carbohydrates that include monosaccharides such as fructose and glucose and disaccharides such as sucrose and lactose [1]. In nature, sugars are present in many plants and fruits because they serve to store energy, primarily from photosynthesis. Many mammals, birds, insects, and bacteria feed on sucrose, fructose, glucose, or other sugars. The heightened preferences for sweets may be related to the physiological properties of sugars. Sugars are vital because they produce the energy necessary for cell functions. They are also required for the quaternary conformation of many molecules essential for life, and they are necessary for the structure and operation of the cell.

Glucose enters the cell by its corresponding transporters to fuel glycolysis. Pyruvate is obtained from this process to enter the mitochondria and forms acetyl coenzyme A, which enters the Krebs cycle. This group of enzymes produces the needed electrons to fuel the respiratory chain, which produces adenosine triphosphate (ATP). One of the biological functions of carbohydrates is to attach to several other molecules and complete their natural process. Metabolically, sugars are the precursors of fat storage.

In mammals, sweetness perception is initiated when sweeteners interact with taste-receptor proteins in taste buds. Then, a signal is generated and translated into communication with the nervous system to identify it. The brain may contain a different group of cells with a food-oscillatory function. Food palatability may shift the neuronal activity from the medial hypothalamus to the limbic and reward-related areas. Complex signaling systems initiate all this communication.

## 2. Gateways through the Membrane

The extracellular environment is not static because many elements make it up, and they are constantly changing so that cells are continuously exposed to a tremendous mixture of environmental states. For its adaptation, the cells must correctly communicate and transmit messages between them. It is only easy to conceive life with compartmentalization. Biological membranes are boundaries between environments, dividing the extracellular, intracellular space, and subcellular organelles. Prokaryotic or eukaryotic cells use lipid membranes, proteinaceous channels, and transporters.

Cellular membranes are dynamic structures with many roles in cellular processes by specific signal-transduction duties, so they can localize, amplify, and direct signals. Their intrinsic properties depend on lipid and protein interrelations. For years, protein interactions have been regarded as the main factor responsible for stabilizing membrane macro- and microdomains. The protein–lipid bilateral exchange has been considered mutually exclusive; nevertheless, they cooperate in creating membrane structural and functional heterogeneity [2].

Membrane proteins are internalized and then degraded or recycled back into the membrane. The actin cytoskeleton regulates membrane dynamics to deform the membrane, promoting invagination, tubulation, and scission of transport carriers in the secretory and endocytic pathways, coordinately holding transport carrier biogenesis and transport.

Membrane transporters are a large and miscellaneous group of proteins characterized by a complicated mesh of receptors, channels, carriers, translocators, and pumps. All of them are considered indispensable. They belong to the major facilitator superfamily (MFS). Solute carriers (SLC) are one of three significant superfamilies of membrane transporters, including uniporters, exchangers, and symporters. SLC proteins are a group of transmembrane proteins that mediate solute in and out flux across membranes. They use substrate and ion gradients to promote substrate transport [2]. They can transport solutes such as glucose, inorganic ions, amino acids, lipids, neurotransmitters, and drugs. 

Another communication system with the outside of the cell is the transmembrane-receptor (TM) proteins that interact with an extracellular molecule, producing an intracellular signal which spreads and amplifies through the canonical molecular transduction pathways and propagates the signal through phosphorylation, activating the signal from downstream. Examples are receptors for insulin, cytokines, neurotransmitters, and sex hormones, to mention some [3]. 

Ion-specific transmembrane channels also sense ion concentrations between the cytoplasm and the extracellular environment. These changes can be monitored by the membrane channels provoking an instant, local response, altering the location and function of membrane proteins. Some examples are the sodium, potassium, and chloride channels, which are very important to maintain life [3].

The Most Recent Common Ancestor (MRCA) is an ancestor of the sugar transport family, which probably goes far back in evolutionary history. This family encompasses transporters through the diversity of life, from humans to bacteria, accepting only sugars or related compounds [4]. 

Sugar transporters (ST) belong to a large family of membrane proteins containing transmembrane α-helices. The oxidation of glucose is the primary source of obtaining cell energy and requires a transport system into the cell, given that glucose cannot diffuse through due to its high molecular weight. Transport is possible through solute carrier proteins; by a process called facilitated diffusion, which takes place down the gradient from high concentration to low concentration. The orchestrated action of various hormones such as glucagon, estradiol, adrenaline, cortisol, thyroid, growth hormone, and insulin achieve glucose normal range levels.

Nowadays, there are diverse transporter proteins in living beings that mediate the entrance of sugars through the lipid layer: Sodium-dependent glucose transporters (SGLT; *SLC5* family), glucose transporters (GLUTs; *SLC2* family), Sugars Will Eventually be Exported Transporter (SWEETs; *SLC50* family), ATP-binding cassette transporters (ABC), fungal sugar facilitators (FfZ family), putative sugar transporters (*SLC45* gene family), (*SLC37* gene family), Nucleotide sugar transporters (*SLC35* gene family), tonoplast monosaccharide transporter (TMT), vacuolar glucose transporter (VGT), sucrose transporters (SUC or SUT) plant transporters, and more.

## 3. SGLTs

The sodium-dependent glucose transporters are integral membrane proteins considered cotransporters or symporters that mediate glucose transport with a much lower affinity to galactose across the plasma membrane by an active transport mechanism. This transport favors the movement of a sodium ion across the plasma membrane into the cell driven by a concentration gradient and a membrane potential and introduces the glucose molecule. Sodium ions cross the apical cell membrane by a sodium/potassium ATPase pump through the basolateral membrane. 

They are composed of 14 α-helical transmembrane regions (TM). The structure contains a central group of seven helices (TM1, TM2, TM3, TM6, TM7, TM8, and TM10) supported by a ring of other helices. Cytoplasmic loops of TM1, TM2, TM5, TM7, and TM9 form a hydrophilic cavity comprising the inner gate of the sugar-binding site. These transporters are of great importance in the physiology of animals, and they are present in mammals’ gastrointestinal tract and renal tubules (Table 1) [5]. Other sodium-dependent sugar transporters may still be waiting to be found in the human genome. NaGLT1, a novel transporter, is a 484-residue protein with 22% amino acid identity with SGLT and GLUT transporters. It has been cloned from a rat kidney cDNA library [6].

## 4. GLUTs

The mammalian monosaccharide transporters or glucose-transport proteins (GLUTs) belong to a family of integral membrane proteins that catalyze the facilitated diffusion (transport down a concentration gradient) of hexose and pentose sugars into and out of cells. They contain 12 hydrophobic, α-helical TM connected by hydrophilic loops. These loops are of different lengths; specifically, they join domains 6 and 7 of the protein. Some GLUTs also mediate the transport of dehydroascorbic acid, urate, or myoinositol. The physiologic substrates for some GLUTs are not known. The GLUTs are expressed in nearly all mammalian cells. However, most cells express one GLUT isoform as the major monosaccharide transport protein and lower levels of one or more isoforms [10].

The human SLC2 protein family contains 14 currently known isoforms, and each member has a distinct tissue-specific expression and binding affinities for individual hexose substrates, considered uniporters. A facilitative glucose-transport system operates in a concentration gradient and is grouped into three classes. These classifications are due to the specific differences in the structure of the proteins (Table 2).

Class 1: transporters 1, 2, 3, 4, and 14; Loop 1–2 is longer than the other five extracellular loops and contains only one glycosylation site. These transporters have an N-linked asparagine with glycan polymers that differ in length, allowing them to move throughout the protein. They express a motif in loops 7 and 8 that confers the possibility of changing conformationally during the sugar transport.

Class 2: transporters 5, 7, 9, and 11; transporters belonging to this class have glycation on the exofacial arginine in the fifth exofacial loop, which connects the ninth and tenth transmembrane spans. This class of Gluts does not have a glycation site on the first loop as the other two classes do. Something that distinguishes class 2 GLUTs is the absence of the tryptophan residue TM10. The transporters from this class prefer fructose over glucose and alternative non-hexose as substrate.

Class 3: transporters 6, 8, 10, 12, and 13 (HMIT); this class of proteins differs structurally from the other two classes. These proteins have an N-linked glycation site on the asparagine in the first loop and a single glycosylation site on an exofacial loop that connects TM 9 and 10, opposite to what happens in classes 1 and 2. They also have intracellular sequences, the kind of dileucine motifs, on the N-terminal domains, which make up part of intracellular targeting and compartmentalization [10,11].

### 4.1. SGLT and GLUT Participation in Diverse Physiological Conditions

#### 4.1.1. Reproductive System

Sugars are essential in all systems, and the reproductive system is no exception. Before reproduction itself, the reproductive cells must be prepared to conduct their work. Therefore, various glucose transporters are present in this kind of cell.

GLUT3 is highly expressed in murine sperm flagellum but absent in bovine sperm, where GLUT5 is the most abundant glucose transporter. This difference shows the concentrations of fructose versus glucose in the seminal fluid of different species. Additionally, GLUT8 is expressed in differentiating spermatocytes and in the acrosome of mouse and mature human spermatozoa [18].

SGLT1 expression and activity in some mammals’ endometrial epithelial cells showed to be responsible for controlling glycogen accumulation essential for embryo implantation. Its activity has recently been reported in normal ovary tissue. This transporter mediates glucose uptake at low extracellular concentrations [19]. Glucose is essential for fetal development, given that the fetus cannot perform gluconeogenesis, which is critical to the placental ability to extract glucose from maternal blood. The sugar is transported to fetal tissues by the GLUTs, and it has been reported that the human placenta expresses GLUT1, 3, 4, 8, 9, 10, and 12.

GLUT1 and 4 have a vital role in glucose regulation, so alterations in these transporters in the communication of mother–fetus glucose transport can have significant consequences for the offspring [20]. GLUT1 was first identified in the endometrium, which increases its expression during gestation, suggesting its participation in maintaining pregnancy. It is a transporter sensible to estradiol, increasing its synthesis due to hormones [21]. Recent evidence supports that insulin enhances glucose uptake in the first trimester but not in the term placenta.

This evidences that GLUT4 and the insulin receptor reduce its presence in the third trimester of gestation. Therefore, the glucose entrance occurs mainly through GLUT1 [20].

GLUT3 is expressed in tissues with a high glucose demand or hypoglycemia using low blood-glucose concentrations, with a low Km for hexoses. It is weakly expressed in the human placenta and only in the early cell-differentiation state. It manifests during the first trimester, when the trophoblast cells are fusing, and not in term placenta, with a similar behavior as GLUT4 [22]. Hypoxic conditions can occur during menstruation and implantation, so GLUT1 and GLUT4 are inducible by hypoxic conditions to promote glucose entrance.

After birth, glucose supply is essential to the lactating mammary gland, providing energy and as a precursor for lactose production. SGLT1 protein is present in lactating mammary gland tissue, but not GLUT1 [6].

#### 4.1.2. Hepatic System

The liver stores and metabolizes fatty acids through β-oxidation in the hepatocytes’ mitochondria, and SGLT2 plays an essential role in fatty acid metabolism. The expression of SGLT2 in humans reveals racial differences [23].

GLUT2 is concomitantly regulated by glucose and a lipogenic factor; the control of GLUT2 transcription thus represents a possible step toward glucolipotoxicity. By controlling GLUT2 transcription, glucose influences metabolism, but GLUT2 is also involved in diet-sugar management.

GLUT9 (*SLC2A9*) has a gene with two different amino-terminal cytoplasmic tails. Human GLUT9a is 540 amino acids long, and GLUT9b is 512 amino acids long. GLUT9b is expressed only in the liver and kidney in humans and mice, while Glut9a is present in many more tissues, including the liver [24]. 

#### 4.1.3. Renal System

In the kidneys, glucose is filtered in the glomeruli and reabsorbed in the proximal tubules through SGLT1 and SGLT2 in the brush-border membrane and maintains a normal glycemia [25]. Glucose reabsorption occurs via glucose transport across the apical membrane by SGLTs and then by passive glucose exit towards the plasma via GLUT2. The low-affinity/high-capacity SGLT2 is responsible for the bulk glucose uptake at the early proximal tubule, sodium/fluid homeostasis, including calcium/phosphate homeostasis, magnesium levels, and glomerular tubular feedback [26], and the high-affinity/low-capacity SGLT1 diminishes glucose to too-low concentrations in the other distal parts of the proximal tubule [25]. Additionally, SGLT1 also serves as a urea and water channel. Unlike SGLT2, which is exclusive to the kidney, SGLT1 is found in the intestine, liver, pancreas, uterus, lungs, eyes, tongue, heart, salivary glands, and prostate [19]. 

SGLT5 (*SLC5A10*) has been reported exclusively in the human kidney cortex transporting mannose, fructose, and glucose [27].

Since fructose itself results in uric acid generation, the observation that GLUT9, a fructose transporter, can also function as a urate transporter raises the possibility that this transporter may regulate the movement of uric acid in and out of the cell in response to fructose [17]. GLUT9 (*SLC2A9*), the isoform expressed in the kidney, analyzes urate transport in this molecule. It is involved in the efflux of urate toward the blood side, and its voltage dependency has recently been confirmed. Mutations in GLUT9 are present in patients with renal hypouricemia [28].

On the other hand, GLUT1 has been detected on the basolateral membrane of the nephron, and GLUT12 is localized in the distal tubules and collecting ducts [29].

#### 4.1.4. Cardiovascular System

Glucose is a significant metabolic heart fuel. An important glucose entrance into the heart exists due to GLUTs. GLUT1 and 4 are principally expressed in skeletal and cardiac muscles and adipocytes. In addition, others have been described, such as GLUT3, GLUT8, GLUT10, GLUT11 and GLUT12. GLUT8 is insulin-sensitive likeGLUT4 and GLUT3, and GLUT10 and GLUT12 are involved in heart development [30].

Besides GLUTs, glucose can be transported by cotransporters of the sodium–glucose cotransporter (SGLT) (*SLC5* gene) family, in which SGLT1 and SMIT1 were shown to be expressed in the heart. SGLT1 has very high expression in the cardiac myocyte sarcolemmal. Van Steenbergen and colleagues proved that testing adult cardiomyocytes with glucose analogs (1-deoxy-glucose, mannose) used by some classical SGLTs does not stimulate ROS production. Interestingly, myoinositol and galactose triggered its production. Thus, SGLT1 and SMIT1 are SGLT isoforms significantly expressed in the heart [9]. 

Some studies have reported the absence of SGLT2 in the heart. In contrast, others suggest that SGLT2 is ubiquitously expressed in most human tissue or present in endothelial cells where SGLT1 and SGLT2 mRNA is overexpressed [31].

SGLT2 might modulate nutrient availability in cardiomyocytes and might influence the cardioprotective effect. However, during stress, such as starvation or hypoxia, SIRT1, a redox-sensitive nicotinamide adenine dinucleotide-dependent enzyme, is activated to maintain the glucose level [26]. Platelets are known to shift their metabolism upon activation, with an increase in glycolysis. GLUT1 and GLUT3 are expressed in platelets, and GLUT3 moves from α-granules to the plasma membrane when degranulated, promoting increased glucose uptake [32]. The function of DHA transport may be involved in maintaining redox levels in the cell or subcellular compartments [11].

#### 4.1.5. Skeletal System

Bone formation and regeneration are costly events. Osteoblasts are essential in bone remodeling and glucose homeostasis because they are the main cellular component of skeletal tissue. Osteoblasts are cells derived from mesenchymal stem cells, and by the influence of runt-related osteoblast transcription factor (*RUNX2*) and osterix, they differentiate into preosteoblasts and eventually into osteoblasts. Osteoblasts primarily rely on glucose as their primary carbon and express three of the four known members of the family of glucose transporters (GLUT1, 3, and 4). Osteoblast differentiation requires GLUT1 and *RUNX2*, where *RUNX2* promotes GLUT1 expression by binding and activating the *Glut1* promotor.

Meanwhile, osterix inhibits chondrogenesis, favoring osteogenesis. GLUT1 and GLUT3 expressions are not very different during all stages of osteoblast differentiation. On the other hand, GLUT4 expression increases insulin dependently, and GLUT2 is not detectable in osteoblasts [33]. GLUT1, GLUT3, and GLUT9 are expressed in healthy articular chondrocytes, promoting chondrocyte physiology and metabolism in cartilage matrices [34].

*SLC37a2* encodes for a sugar transporter critical for bone metabolism and highlights the previously unappreciated plasticity of the osteoclasts’ specialized lysosome-related organelle(s) and can be a potential therapeutic target for metabolic bone diseases [15].

#### 4.1.6. Immune System

The immune system is essential to defend against infectious organisms and strange products. It is characterized by different cell types: neutrophils, eosinophils, basophils, thymocytes, lymphocytes, and monocytes (monocytes differentiate into macrophages when they migrate from the blood into organ systems) [10]. Bone marrow stem cells possess GLUT5 and regulate GLUT3 in the presence of glucose variation, and this occurs just before they are differentiated and thrown out into the bloodstream [34]. The erythroid lineage expresses GLUT1 predominantly.

The major glucose transporter in lymphocytes is Glut1, but GLUT3, GLUT4, GLUT5, and GLUT6 are expressed in these cells. Lymphocytes divide rapidly and need glucose to do it. GLUT1, GLUT4, and GLUT3 transporters mediate thymocyte glucose uptake, and in splenocytes, the transport is due to GLUT1 and GLUT4 [10]. GLUT1, GLUT3, and GLUT4 are expressed in resting monocytes and polymorphonuclear cells.

The differentiation of monocytes into macrophages overexpresses GLUT3 and GLUT5, and the expression of GLUT1, GLUT3, and GLUT5 must be maintained through the transformation to foam cells [35].

GLUT6 has the potential to modulate the glycolysis pathway in inflammatory macrophages.

Insulin is known to modulate the immune response either directly or indirectly by modifying basal and proliferative responses of immune cells, so during a metabolic disturbance, as in the metabolic syndrome, the immune system is constantly activated and responding to infections by regulating glucose transport by activating different GLUTs in different immune-cell populations [36].

As for the SGLTs, SGLT6 (*SLC5A11*), now known as SMIT2, has the lowest amino-acid identity of the SGLTs, but there is evidence that *SLC5A11*, SGLT6, or SMIT2 interact with immune-related gene(s) and may function as an autoimmune modifier gene in humans [25].

#### 4.1.7. Nervous System

The primary sugar for neurons is glucose, which requires active glucose transporters such as GLUT and SGLT [37]. SGLT1 is expressed not only in the intestine and kidney but also in specific brain regions, such as the CA1cells of the hippocampus and the Purkinje cells in the cerebellum, where it may act as a glucose, sodium, urea, or water transporter. In humans, this protein is expressed in cholinergic neurons in the enteric nervous system and at the neuromuscular junction. Rat hypothalamic neurons express both SGLT1 and SGLT3a, where they may play a role in glucose sensing by glucose-excited neurons [28]. SGLT2 expression is shallow in the brain but rises in patients with traumatic brain injury [24].

SGLT1 and 2 are associated with learning, food regulation, and energy homeostasis consumption [38].

Glucose is transported from the circulation through the endothelium to glial cells by GLUT1. This transporter plays a vital role in circulating glucose from blood to the brain parenchyma through the brain–blood barrier (BBB) [39]. Choroid plexus (CP) epithelial cells also use GLUT1 as the primary glucose transporter [40]. 

GLUT1 is related to neuronal maturation and stability, and GLUT3 is also directly associated with the increased expression of specific proteins, such as synaptophysin (SNAP-25), and the α3 subunit of Na^+^-K^+^-ATPase, which correlates with the maturation and regional cerebral glucose utilization (rCGU) profile [41].

GLUT3 is the central mediator of glucose uptake into neurons, which is in axons and dendrites, with a higher glucose affinity and transport capacity to ensure the preferential use of glucose by neurons [42]. GLUT4 is also expressed in a subset of neurons, especially in cholinergic neurons of the rat forebrain, often coexpressed with GLUT3. GLUT4 may function to rapidly increase glucose uptake into specific neurons in response to increased energy demand. During insulin resistance, there is a decline in memory, where glucose transporters GLUT1 and GLUT3 could be involved [43]. 

In the brain, GLUT8 is found in the hippocampus, dentate gyrus, amygdala and primary olfactory cortex, hypothalamic nuclei, and the nucleus of the tractus solitarius. High GLUT8 levels are present in the supraoptic-hypophyseal tract, localized to synaptic vesicles and vasopressin-containing secretory granules in the posterior pituitary.

GLUT12 is expressed in human and mouse brains, and its role in glucose homeostasis under physiological or pathological conditions remains unclear. 

HMIT is an H^+^/myo-inositol cotransporter (GLUT13), and its transport can be inhibited by phloretin, phlorizin, and cytochalasin B, but no glucose transport can be observed. It is expressed predominantly in the brain, hippocampus, hypothalamus, cerebellum, and brainstem. Additionally, its translocation to the plasma membrane occurs at growth cones and synapses triggered by neuronal activation and increased Ca^2+^ entrance [17].

Some neurons in the ventrolateral preoptic nucleus (VLPO) participate in the induction and maintenance of sleep waves. Some neuronal populations can regulate glucose homeostasis or feeding behavior, and these cells increase their firing activity due to a rise in extracellular glucose. 

Some studies have demonstrated that sleep is closely related to metabolism and its substrate compounds, such as glucose, glycogen, or lactate. ATP synthesis in these neurons is directly related to glucose metabolism. A rise in ATP closes the K^+^/ATP channels and leads to cellular depolarization, and its levels fluctuate during vigilance states. Hence, glucose contributes to sleep onset. VLPO neurons express glucose-transporter GLUT3 mRNAs and the enzyme glucokinase [33].

### 4.2. SGLT and GLUT Participation in Disease

#### 4.2.1. Obesity

Obesity is a growing health problem characterized by excessive adipose tissue accumulation. This condition increases cardiovascular disease, neurodegenerative disease, osteoarthritis, peripheral vascular disease, cancer, renal failure, and Type 2 diabetes mellitus (T2DM). White adipose tissue expresses some glucose transporters such as GLUT-1, GLUT-4, GLUT-5, GLUT-8, GLUT-10, GLUT-12, and HMIT [44]. 

This condition causes an imbalance in the metabolism of carbohydrates and lipids. Within its decompensation is a differential expression of the glucose transporters. 

Decreased GLUT-5 expression has been reported in the intestine of subjects with obesity [45]. In contrast, other discoveries have determined that obesity displays enhanced duodenal SGLT1 and GLUT-5 abundance. This last discovery could explain the increase in blood glucose in the postprandial period, insulin resistance, and hyperinsulinemia [46].

Other obesity studies also found an altered expression of GLUTs in the liver, fat, and muscle. GLUT 1 and 3 were upregulated, while GLUT 2 and 4 were downregulated. This condition could be explained by a relation between class 1 GLUTs and metabolic disorders in overweight patients [47]. The decrease in GLUT4 has also been observed in the adipose tissue of people with morbid obesity, causing insulin resistance [48].

Another affected transporter is GLUT 12, which is decreased in tissues that regulate carbohydrate metabolism [49].

Being overweight also impairs glucose homeostasis associated with intestinal and renal glucose-altered absorption due to SGLTs. The physiological function of SGLT1 is its role in oral rehydration therapy used to treat infectious diarrhea. These transporters act as channels for water and small hydrophilic solutes [6]. In morbidly obese individuals, glucose absorption in the proximal small intestine is attributed to increased SGLT1. SGLT3 is a homolog of SGLT1 that acts as a glucose sensor in the gut and portal vein. Intestinal SGLT3 expression is localized in the epithelium, preferably expressed in the small intestine, though absent in the colon. (It is noteworthy that human, rat, and mouse SGLT3 do not transport glucose, but in oocytes can generate currents by membrane depolarization due to glucose or sodium exposure.

Interestingly, SGLT3 can also provoke ion currents without sodium and proton transport but with glucose. Based on the idea that it is a sensor, SGLT3 expression may be altered in obesity. Additionally, SGLT3 expression in obese patients is lower compared with lean individuals, and when obese people undergo bariatric surgery, SGLT3 rises [23].

*SCL5A11* (SGLT6) transports myoinositol and D-glucose to a lesser degree in a sodium-dependent manner. It is strongly expressed in the small intestine and brain areas linked to the control of reward processing and, hence, food intake, such as the hypothalamus and *substantia nigra*. These areas are related to the food intake control and reward circuit, and they may play a role in recognizing nutrients and their ingestion. Therefore, it is necessary to investigate its participation in obesity [50]. Being obese produces several adipokines and affects bone metabolism.

#### 4.2.2. Diabetes

T2DM is a chronic degenerative disease characterized by a high blood glucose concentration. Hyperglycemia (HG) harms cardiomyocytes, such as the alteration in myofibrillar structure and intercellular connections. It has been described that glucotoxicity induces reactive oxygen species (ROS) production, causing severe cardiac damage and other cellular and organic damage. A clinical characteristic of T2DM is insulin resistance. A slow blood glucose clearance produces this condition by impaired glucose transport into muscle and adipocytes. In this disease, insulin cannot reduce blood glucose and free-fatty-acid levels, which contributes to more insulin resistance in muscles and the liver [51].

Hyperglycemia affects the manifestation of GLUTs in some tissues. GLUT2 is present in the kidney proximal tubules and increases under diabetic conditions, while GLUT1, GLUT2, and GLUT3 decrease [25]. 

Muscles and adipocytes are critical participants in metabolic control, and the primary glucose transporters present in these tissues are GLUT1, 2, and 4. GLUT4 is not static on the cell surface when insulin mobilizes; it is continuously recycled and stored in vesicles to respond to subsequent insulin stimulation [52]. GLUT4 is the main effector of the insulin signal, and the insulin resistance, characteristic of T2DM, is due to the deteriorated trafficking of GLUT4 into the cell membrane, which contributes to insulin resistance in muscle and adipocytes [52]. GLUT5 levels are significantly upregulated in the skeletal muscle and intestine of type 2 diabetic patients [17].

GLUT8 is also present on the cardiac atrial cell membrane and is also regulated by insulin. During T2DM, its presence is downregulated by up to 90%. These findings can contribute to understanding the pathophysiology of diabetes and its cardiovascular complications [53]. 

Hence, HG-induced ROS production depends on these SGLT isoforms that take advantage of the sodium gradient to move some sugars and not by the classical facilitated glucose transporters, as 2-deoxy-glucose does [8]. In diabetes, intestinal SGLT1 transporters are enhanced by the glucose absorption in this disease, exacerbating the HG. On the other hand, the presence of SGLT2 is decreased in the kidney in response to hyperglycemia. These two transporters have opposite functions during the disease [54]. 

SGLT1 expression is increased in T2DM and ischemia but decreased in T1DM. SGLT1 is at least partially responsible for increased cardiac glucose uptake following exposure to insulin and leptin, and leptin acts by directly increasing SGLT1 expression [55]. SGLT1 expression is altered in diabetic and ischemic cardiomyopathy; SGLT1 expression may be partially regulated by leptin, and SGLT1 mediates at least part of the increased cardiac glucose uptake in response to insulin and leptin.

Evidence suggests that mutations in the SLC5A9 gene (SGLT4, which handles mannose homeostasis) cause serum mannose and 1,5 anhydroglucitol imbalances in T2DM. In this disease, a characteristic symptom is the glucose leak into the urine, overwhelming the SGLTs’ reabsorption capacity in the kidney. Therefore, there are alternative therapies to manage diabetic patients to control blood glucose by inhibiting renal SGLT2 [6]. Its malfunction leads to decreased blood glucose due to renal excretion, as in patients with primary glucosuria [56]. In hyperglycemia, GLUT-9 is the primary uric acid transporter at the basolateral side of the kidney proximal convoluted tubules (PT), and increases with URAT-1. A direct association between SGLT2 and URAT or GLUT-9 expression is still being determined, and new research must be conducted to elucidate the relationship between these transporters [57].

Diabetes suppresses some of the master genes involved in the generation of osteoblasts. These cells are responsible for bone formation in the bone marrow, and an increase in fat in this niche harms bone health. Hyperglycemia favors the formation of advanced glycation end-products (AGEs) by increasing the oxidative with a detrimental effect on the skeleton [58]. 

#### 4.2.3. Inflammation and Immunodeficiency

Inflammation is a process where the organism defeats foreign organisms or molecules by different type of cells and molecular systems.

Hyperglycemia increases inflammation during atherosclerosis’s pathogenesis. SGLT2 inhibitors reduce cardiac events, suggesting that SGLT2 may be involved in inflammation and oxidative stress in cardiovascular events [57,59].

Evidence shows that increased SGLT1 in the intestine can protect against enteric infections, suggesting that this transporter may have a novel immunological role [6].

During hypoxia and oxidative stress, SIRT-1 is associated with its interaction with HIF-1α, which is the principal molecule that governs the inflammatory process in cardiomyocytes [57].

During an inflammatory state, interleukin 1β (IL-1β) overregulates GLUT1 expression. On the other hand, GLUT5 is the only transporter in microglia cells, while GLUT4, like GLUT8, is expressed in the hippocampus and amygdala, where it plays an essential role in glucose homeostasis. In autoimmune diseases, an increase in fibroblasts in the hippocampus and the amygdala transform into macrophages, causing inflammation and the deregulation of glucose transport by changes in GLUT4 and GLUT8 [43].

When the immune system does not recognize the body itself, it causes diseases designated as autoimmune. Regarding SMIT1, a mutation in SLC5A3 has been identified as a progressive anemia in ponies called fatal Foal Immunodeficiency Syndrome [28]. 

#### 4.2.4. Infertility and Miscarriage

SGLT1 deficiency decreases endometrial glycogen and litter size, leading to early pregnancy failure and obstetrical complications, including low fetal growth [20]. 

A possible intervention of GLUTs in uterine pathologies, such as infertility, polycystic ovary syndrome, and endometrial cancer, is proposed. It has been identified that women with idiopathic infertility have lower levels of GLUT1 expression and aberrant expression of GLUT3 and 8 [21]. 

#### 4.2.5. Hypertension

Hypertension is one of the most common medical conditions, and it is caused by a persistent elevation in arterial pressure. Salt can induce hypertension and upregulate the inflammatory factor, transforming growth factor β1 (TGF-β1). Increased filtered glucose enhances SGLT2 activity, worsens glycemic control, and promotes Na+ loading with subsequent impaired blood-pressure control.

GLUTs mediate renal tubular glucose reabsorption and, in diabetic kidneys, are upregulated. GLUT1 and GLUT12 levels are elevated in hypertension and nephropathy. Hyperglycemia and renin-angiotensin system activation are believed to be involved in hypertension [29]. 

Nocturnal hypotension is commonly associated with T2DM patients and may be regulated by SGLT2 [27].

GLUT-1 is ubiquitously expressed, but with SGLT2, both are the predominant glucose transporters in mesangial cells in the kidney. Hypertension produces TGF-β1, which upregulates GLUT-1 expression, and this overexpression increases glucose transport with excess fibronectin and collagen production [60]. Hypertensive animals exhibited a profoundly reduced level of the GLUT4 protein [61].

#### 4.2.6. Metabolic Disturbances

Some mutations in the GLUT2 gene (*SLC2A2*) are responsible for the Fanconi–Bickel syndrome, an autosomal recessive disorder in carbohydrate metabolism. For some of the identified GLUT2 mutations, the transporter function is abolished. Patients with this rare disorder suffer from hepatomegaly, nephropathy, fasting hypoglycemia, sugar intolerance, and growth delays. These patients do not tolerate simple sugars in their diet, and a considerable glycogen accumulation is found in GLUT2-expressing tissues [62].

#### 4.2.7. Cancer

A cancer environment is characterized as acidic and hypoxic. Tumor metabolic plasticity is not only due to the glycolytic phenotype (Warburg effect), but also mitochondrial energy reprogramming. These two metabolic routes are necessary for a high energy demand and biomolecule precursors so the cancer cell can grow.

Glucose metabolism alterations are characterized by an elevated glucose uptake, hyperactivated glycolysis, decreased oxidative phosphorylation component, and lactate accumulation. The key enzymes, such as hexokinase, lactate dehydrogenase, and enolase, are upregulated and activate the transporters of glucose metabolism. Drug elimination comprises two energy-dependent mechanisms contributing to malignant cells’ resistance to chemotherapy.

Besides using aerobic glycolysis as a rescue system to obtain energy, cancer cells use an immune-system-evading strategy. A metabolic competition exists between immune and malignant cells during nutrient deprivation, mainly glucose [63]. Therefore, the only way to introduce glucose to the cell is through the GLUTs. The overexpression of GLUTs is related to aggressive tumors and poor survival prognosis, implying that tumors with significant glucose uptake that are metabolically more active are more aggressive. This phenomenon is often associated with the deregulated expression of GLUT1 and GLUT3 in all kinds of tumors. These two transporters share some regulation mechanisms in response to HIF-1α or p53 and usually respond to different stimuli. Transcription factors *c-Myc* (encoded by the well-known MYC oncogene) and *sine oculis homeobox 1* (SIX1) stimulate glycolysis directly by GLUT1 transactivation, for example [64]. 

Furthermore, the mammalian target of the rapamycin (mTOR) pathway upregulates GLUT3, and a link between oncogenic EGFR signaling and GLUT3 has also been demonstrated in lung adenocarcinoma. The expression of GLUT3 occurs through membrane localization during immune cell activation, which is relevant for the tumor microenvironment [64]. GLUT3 mRNA and GLUT5 mRNA expression are similar in primary tumors compared to normal tissue, whereas their expression is higher in metastases [19].

GLUT1 is detected in atypical hyperplasia and endometrial adenocarcinoma compared to a healthy state, as shown in many malignant tumors. Additionally, there is an increased expression of GLUT8 in endometrial cancer [21].

Osteosarcoma is one of the most common pediatric malignancies of bone and soft tissues. Like other cancers, this type also has a high glucose uptake. Glut-1 expression was upregulated in osteosarcoma tissues compared with adjacent non-cancerous tissues [65].

Many GLUTs are involved in this permanence tactic and differ between tissues and cancer types, as seen in Table 3.

SGLT1 is expressed in primary tumors and metastatic lesions of the lung, pancreatic adenocarcinomas, and head and neck cancers, and is also present in colorectal and prostate tumors; meanwhile, SGLT2 is expressed in metastatic lesions of lung cancers, in colorectal, gastrointestinal, head and neck, and kidney tumors, as well as in chondrosarcomas and leukemia. Only one study describes SGLT2 overexpression in ovarian cancer [20]. SGLT2 has been observed in human hepatoma cells, given that they use this glucose transporter to uptake the necessary glucose for its function. However, SGLT2 also exists in liver mitochondria in this cell lineage [27].

It has been possible to observe the role of SGLT1 in cancer cell survival, where the epidermal growth factor receptor stabilizes the transporter and prevents autophagy. Thus, this could explain the tumor cells’ resistance to chemotherapeutic agents [6].

#### 4.2.8. Epilepsy

Epilepsy is a brain instability characterized by repeated electrical activity [73]. Some studies have been carried out in the search for glucose transporters, but only an association of Glut1 has been found with the various manifestations of epilepsy. Glut1 is the primary carrier of glucose in the blood–brain barrier, and the nervous system is not an exception to problems in glucose transport. A dysfunctional GLUT1 transporter causes infantile epilepsy, designated GLUT1-deficiency syndrome (GLUT1DS). This illness causes some clinical phenotypes, such as severe mental delays, learning disabilities, delayed development, infantile seizures, ataxia, and altered glucose utilization in the brain. This disease also provokes abnormal sleep–wake patterns [74]. On the other hand, mutations in *SLC2A1*, which codes for Glut1 and is associated with severe epileptic encephalopathy [74,75]. 

#### 4.2.9. Alzheimer Disease

Alzheimer’s is a degenerative brain disease which slowly destroys memory and thinking ability. Areas or patterns of reduced glucose metabolism are often seen in brain scans of patients with Alzheimer’s and other dementias. A growing body of evidence suggests that glucose hypometabolism may be more than just a biomarker on brain scans.

GLUT12 expression was upregulated in the frontal cortex of patients with Alzheimer’s disease (AD) [37]. AD patients show diminished GLUT1 and 3 levels, especially in the cerebral cortex, with significant loss of GLUT3. One such study has also demonstrated the reduced expression of GLUT3 in the dentate gyrus. At the same time, another reported a considerable lowering of GLUT1 expression with no difference in mRNA levels of GLUT1 in the human AD brain, suggesting post-transcriptional regulation [76]. In AD, it was observed that high glucose concentrations in the brain are associated with greater severity in the deposition of amyloid plaque, associated with a decrease in the expression of GLUT3, but not GLUT1 [77].

#### 4.2.10. Aging

Aging results from the impact of the accumulation of various types of molecular and cellular damage over time. A common comorbidity observed in aging is metabolic dysfunction. Glucose and mitochondrial dysfunction are frequently associated with aging, but the causal relationship between aging and metabolic dysfunction is not fully understood.

Autonomous system settings are impaired as we age, so older people are more vulnerable and frail. Recently, it has been possible to relate the new diabetic agents, such as SGLT-2 inhibitors which can cause falls in geriatric patients, to age, suggesting the participation of these transporters in regulating equilibrium [78].

Usually, glucose reabsorption is mainly mediated by SGLT2, and under SGLT2 malfunction (as mutations), the contribution of SGLT1 increases. Blood glucose levels significantly decrease age-dependently, which can lead to thinking that such compensation of SGLT1 is altered with aging [26].

Another fact is that aging affects insulin resistance and the development of T2DM, as control of the blood glucose level declines and glucose disposal deteriorates with time. GLUT1 expression declines after birth, but its muscle localization remains unclear. In skeletal muscle, insulin and exercise trigger the translocation of GLUT4 to the membrane, promoting glucose uptake. GLUT4 expression has a significant correlation with age. Muscle contraction needs an increase in glucose uptake, but age does not influence this process. However, glucose uptake under basal conditions decreases with age [79]. 

Senescent neurons exhibit glucose metabolism dysfunctions. Among them are insulin resistance, glucose transport perturbation, and mitochondrial dysfunction. Other cells which are closely related to glucose metabolism are the microglia. These cells express all the key enzymes of glucose metabolism and GLUT1 and GLUT3 and display manifestations related to senescence during aging [41].

#### 4.2.11. Sleep Disturbances

Long-term sleep deprivation can cause multiorgan failure and neurodegenerative disorders. Insomnia is a sleep disorder and a prevalent health concern [80].

Kordestani et al. discovered that GLUT1 expression increases in the amygdala during sleep deprivation, so it can be used as a metabolic biomarker [81]. When sleep is disturbed by obstructive apnea, it has been possible to determine an increase in GLUT4, with its associated insulin resistance. In addition, increases in inflammatory markers such as Tumor necrosis factor α (TNF-α) have been recorded [82]. 

#### 4.2.12. Osteoporosis

Osteoporosis is classified as a bone atrophy with an elevated risk of fracture, especially in elderly persons. It manifests by an asymptomatic decrease in bone mineral density. GLUT1 is primarily involved in bone formation, specifically in osteoblast differentiation, and plays a role in supporting the regular mineralizing activity of osteoblasts. A defect in the presence of GLUT1 may be involved in bone formation deficits [83].

Patients with diabetes have a decreased bone mineral density and therefore increased risk of fractures. However, when SGLT inhibitors control blood glucose levels, some reports indicate a detriment to bone health, causing possible fractures [84].

## 5. SWEETs

SWEETs are sugar-mobilization systems containing passive facilitators and sugar/H+ symporters present in bacteria, plants, fungi, and only one human cell lineage. The founding member of this family is *MtN3*, first identified in the legume *Medicago truncatula*, then a homolog of *MtN3* was identified in *Drosophila melanogaster* and named Saliva; the *MtN3*/Saliva motif is present in many organisms, from bacteria to eukaryotes. Recently, they have demonstrated their sugar-transporter function and been given the name SWEETs.

They are passive facilitator sugar transporters, allowing sugars to enter through their concentration gradient across the cell membrane. Meanwhile, symporters do this against their concentration gradient by a coupling transport process using a co-substrate that follows its concentration gradient [85]. They are conserved from archaebacteria to plants and humans. 

SWEETs appear to play critical roles in the efflux component of cell-to-cell transport and the secretion of sugars. Seventeen types of SWEETs have been described in plants that serve distinct physiological functions. SWEETs are vital because they are targets to be studied to boost crops and increase nutrient content and stress resistance to plant salinity and pathogens. Seven SWEETs have also been identified in fungi [85], absent in the Ascomycota species, which use polyol transporter proteins [86].

SemiSWEETs have been found in bacteria, and several members of this family transport sugars. They are built from only seven TM-spanning domains. These sugar symporters’ co-substrates are protons, compared to the sodium ions used by many other sugar symporters (Table 4) [85]. They are organized into four groups; clades I, II, and IV are predominantly hexose transporters. Meanwhile, clade III is predominantly a sucrose transporter, although it also can transport hexoses [87].

Glucose induces the translocation of the transient receptor potential vanilloid (TRPV2) to the plasma membrane. Overexpression of HsSWEET1, also called recombination activating gene 1 activating protein 1 (*RAG1AP1*), may lead to altered glucose levels, affecting TRPV2 targeting. HsSWEET1, expressed in β-cells, may contribute to glucose homeostasis and, thus, like GLUT2, may play a critical role in glucose-induced insulin secretion. Given the importance of sugar homeostasis in humans, it is necessary to study the physiological role of SWEETs and their regulation, including metabolic diseases [89]. The expression of mouse MmSWEET in the mammary gland was suggestive of a role in Golgi lactose synthesis [90].

*Caenorhabditis elegans* has a homolog CeSWEET1 (*Swt-1*) that transports glucose and trehalose, playing an essential physiological role in this worm [91]. Many classifications are due to the species in which they are found.

Carbohydrate partitioning assimilates, transports, and distributes sugars to produce plant growth and development. Sugar transport is crucial in the cell–cell and long-distance sugar flux in plants. In seeds, the lipids and proteins are converted to sucrose. Due to the high-sucrose concentration in the conduction tissue in plants, its exit does not require energy. Therefore, the sugar transit is produced by SWEETs or H+/sucrose cotransporters functioning as effluxers, sucrose transporters, or sucrose carriers (SUC/SUT). Sucrose fate depends on the presence of some enzymes called invertases (CWI), so the sucrose is cleaved to hexoses and then taken by monosaccharide transporter (MTPs), ST, or SWEETs (Table 5). SWEETs seem to export sucrose and monosaccharides and can facilitate the influx and efflux of sugars [92]. Their nomenclature is assigned for both SWEETs and semiSWEETs, depending on the specific organism in which they are found. Each species can have more than one transporter type, so the terminology becomes very complex.

RiMST5 and RiMST6 are two novel high-affinity monosaccharide transporters from *Rhizophagus irregularis* that modulate the plant–fungus close interactions [93].

The final entrance step occurs with either SUC4 or hexose transporters (VGT, vacuolar glucose transporter; TMT1 and 2, tonoplast monosaccharide transporter, SWEET2). This review will not examine these pathways and the transporters involved in plants; nevertheless, they can be consulted in [92,94,95,96], to name a few.

In fungi, there are 49 unique fungal sugar facilitators called Ffz, previously shown to be required for yeast’s fructophilic metabolism. Fructose may be preferred in W/S-clade species because it can be converted directly to mannitol, with an impact on redox balance. They do not belong to the major facilitator superfamily. Ffz1 is a high-capacity and low-affinity uniporter, specific for fructose. It is a prerequisite for fructophily in at least one yeast species, while Ffz2 transports glucose and fructose. Ffz transporters have been found only in a limited number of fungal species [4].

## 6. ABCs

ABC transporters are primary active systems that use the energy derived from ATP hydrolysis to transport substrates against a concentration gradient. They have a prototypical structure containing two cytoplasmic nucleotide-binding domains (*NBD1* and *NBD2*) and two transmembrane domains (TMD1 and TMD2), each formed by six membrane-spanning α helices. All eukaryotic ABC transporters are effluxers, and so far, no human or plant ABC transporter is known to transport sugars. Meanwhile, bacterial ABC transporters can transport maltose, which, upon ATP hydrolysis, adopts the inward-facing open state with the release of maltose to the cytoplasmic side like GLUT1, in which residues responsible for sugar binding are located in only the C-domain half of the translocation pore [99]. 

All bacteria, commensals, and pathogens establish colonization in the host by nutrient uptake. Pathogenic bacteria can rapidly adapt to changing microenvironments through nutrient acquisition by select ABC transporters. The mechanisms to acquire those nutrients are essential for its virulence used to mediate disease. ABC importers are further divided into three categories; Type I, Type II, and Type III transporters (also known as energy-coupling factor (ECF) transporters) [100]. However, Thomas and colleagues suggest a new ABC transporter classification [101].

Bacterial ABC transporters that mediate uptake utilize a high-affinity solute-binding protein located in the periplasm of gram-negative bacteria and can be tethered to the cell surface or incorporated into the transporter itself. These binding-protein-dependent transporters take up various substrates, including nutrients and osmoprotectants including small sugars, amino acids, peptides, metals, anions, iron chelators, and vitamin B12. The class 1 and class 2 binding proteins, represented by the galactose-/glucose-binding protein and the maltose-binding protein, respectively, differ primarily in the fold of the two domains [100]. PglK is an oligosaccharide ABC transporter from a bacterial protein N-glycosylated system [102]. 

ABC transporters can also efflux substances from bacteria, such as bacterial cell components (such as capsular polysaccharides, lipopolysaccharides, and teichoic acid), proteins involved in pathogenesis (such as hemolysin, heme-binding protein, and alkaline protease), peptide antibiotics, heme, and drugs. 

The E.c.MalK component of the maltose transporter from *E. coli* belongs to a group of bacterial sugar transporters. The intact maltose transporter using vanadate has provided compelling evidence that the ATP-sandwich dimer represents the physiological conformation in the entire transporter [103]. The loss of a selective substrate’s transport results in an inability of the bacterium or pathogens to internalize nutrients and support pathogenesis. Targeting these acquisition nutrient systems may provide therapeutic alternatives to disrupt bacterial colonization [100]. 

## 7. Other Transporters

In recent years, a new putative sugar transporter family (*SLC45* gene family) has been proposed, with four proteins identified. Considering their structural similarity with the plant and Drosophila melanogaster sucrose transporters, it can be suggested that there is a phylogenetic relationship with corresponding proteins from plants, fungi, and bacteria. *SLC45A1* is an H+/sugar symporter that regulates neuronal glucose uptake under induced acid stress. *SLC45A2* has an essential role in melanin synthesis and seems intrinsic to melanocytes, but it is also present in muscles, ovaries, testes, kidneys, and uterus. It is similar to plant H+/sucrose-symporters, so it is proposed to regulate cellular osmolarity. In addition to its putative role as a sugar transporter, recent studies have demonstrated the role of *SLC45A3* in prostate cancer genesis related to the erythroblast-transformation-specific (ETS) family of transcription factors. *SLC45A4* has been found in moderate amounts in some human organs; it is absent in others, with no apparent function described until now [104].

The Spinster (*SLC63A1*) gene encodes a mammal homolog, *Spsn1*, responsible for the lysosomes’ sugar export [18].

Protein glycosylation is the most prevalent and complex post-translational modification, and this process attaches sugars to the growing end of glycan chains in the new proteins. The SLC35 family is a group nucleotide sugar transporters (NSTs) that connects the synthesis of activated sugars in the nucleus or cytosol to glycosyltransferases that reside in the lumen of the endoplasmic reticulum (ER) and/or Golgi apparatus. *SLC35A1* deficiency has been referred to as a congenital disorder of N-linked glycosylation IIf (CDG-IIf). This abnormal glycosylation is crucial in diabetes, cancer, inflammatory disease, neurodegenerative processes, and neuromuscular disease [105].

## 8. Economic Importance of Sugar Transporters

Some of these MFS families function as efflux pumps to extrude various toxic materials from cells and contribute to drug resistance and stress tolerance.

Yeasts have not only evolved to grow efficiently on simple sugars in nature but have also been accompanying human civilization and are very important in many fermentation processes. Knowing how these microorganisms introduce their sugar substrate has permitted their domestication. On the other hand, there are opportunistic fungal pathogens of humans and animals causing diseases. Therefore, studying their way of surviving is a method of protection against them [106].

## 9. Targeting Glucose Transporters in Clinics

Knowing its structure and molecular characteristics is essential to synthesize or search for a compound that can neutralize another. The clinic has advanced thanks to the use of compounds, synthetic or of natural origin, that can inhibit pathways in which glucose transporters are involved.

The overexpression of GLUT has been clinically exploited in tumor imaging via fluorodeoxyglucose (FDG) positron emission tomography (PET). GLUT differential expression in cancer and immune cells is used as a biomarker to uncover immune functionality in cancer patients and those who underwent immunotherapy [62].

GLUTs involved in cancer can be inhibited by several compounds (WZBs, polyphenols) or molecular techniques (miRNAs, shRNAs, siRNA, and antisense cDNAs). Another way to disable these transporters is to create conjugates with toxic compounds to target specific cell populations (Figure 1) [107].

The hypoxic microenvironment in some autoimmune diseases upregulates the GLUT1 expression by either T-cell-specific knockouts or small-molecule GLUT1/glycolysis inhibitors, by which improvement can be seen in arthritis rheumatoid, lupus, or psoriasis [108].

Combining metabolic inhibitors, such as 2-deoxy-D-glucose (2-DG) and metformin, has given promising results in preventing chemotherapy resistance [109].

SGLT2 inhibitors have been very useful in controlling high pressure and protecting the heart [29] (Figure 1).

The inhibition of transporters can also be addressed by manipulating transcription factors or intervening in the mechanism that transports them to the membrane. 

## 10. Conclusions

Sugar transport is highly regulated and expressed differentially in tissues and different organisms depending on the function they control. Until now, various aspects of these transporters have been characterized in detail, such as the distribution in tissues, its specificity to the substrate, its kinetics, and in the case of some, their physiological role. However, there is still much to know, such as mechanisms that regulate its synthesis, incorporation into intracellular vesicles, translocation, internalization, degradation, etc. The detailed knowledge of these systems of transport and their regulation in the future will allow us to design more efficient therapeutic strategies in the case of their dysfunction.

## Figures and Tables

**Figure 1 ijms-24-06085-f001:**
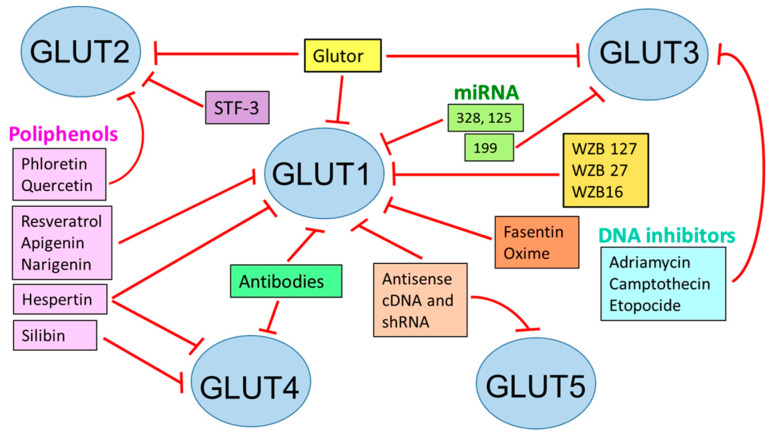
Inhibition of glucose transporters for therapeutic effects. STF: specific transcription factor 3; WZB: small molecule GLUT1 inhibitor; Based on [107].

**Table 1 ijms-24-06085-t001:** SLC5 family nomenclature, substrates, and expression.

**Transporter**	**Transports**	**Km**	**Localization**	**Ref.**
SGLT1 *(SLC5A1)*	1 Glucose/1 Galactose:2 Na^+^ αMG	0.5/1 mM	intestine, trachea, kidney, heart, and colon	[7,8]
SGLT2 (*SLC5A2*)	1 Glucose:1 Na^+^ αMG	5 mM	heart and kidney cortex	[7,8]
SGLT3 (*SLC5A4*)	1 Glucose:2 Na^+^ imino sugars αMG	20 mM	glucose sensor in the enteric nervous system, uterus, lung, brain and autonomic nervous system, spleen, thyroid, kidney, and trachea	[6,7,8]
SGLT4 (*SLC5A9*)	Mannose, fructose, and glucose αMG	2 mM	small intestine, brain, and kidney	[6,8]
SGLT5 (*SLC5A10*)	Mannose, fructose, and glucose αMG	ND	kidney cortex	[6]
SMIT1 (*SLC5A3*)	Myo-inositol:Na^+^, 1 Glucose/1 Galactose: Na^+^ αMG	0.05 mM	heart, brain, and kidney	[9]
SGLT6 or SMIT2 (*SLC5A11*)	Myo-inositol: Na^+^ αMG, xylose and glucose	0.3 mM	spinal cord, kidney, and brain	[6,8]

αMG: α-methyl-D-glucopyranoside; ND: not described.

**Table 2 ijms-24-06085-t002:** SLC2 family nomenclature, substrates, and expression.

Transporter	Transports	Km	Localization	Ref.
CLASS I				
GLUT1 (*SLC2A1*)	Glucose, mannose, glucosamine, DHA, fucose, and galactose	2 mM	erythrocytes, brain endothelial cells, neurons, kidney, heart, lymphocytes, mainly all cells	[12,13,14,15,16]
GLUT2 (*SLC2A2*)	Glucose, mannose, glucosamine, fructose, and galactose	17 mM	β-pancreatic cells, liver, brain, kidney, small intestine	[8,12,13]
GLUT3 (*SLC2A3*)	Glucose, mannose, DHA, xylose, and galactose	2 mM	central nervous system, placenta, liver, kidney, heart, lymphocytes	[12,13,14]
GLUT4 (*SLC2A4*)	Glucose, DHA, and glucosamine	5 mM	insulin-sensitive tissues, heart, central nervous system, heart, lymphocytes	[8,12,13,14,16]
GLUT14 (*SLC2A14*)	Glucose	ND	testis	[13]
CLASS II				
GLUT5 (*SLC2A5*)	Fructose	10 mM	small intestine, testis, kidney, central nervous system	[8,12,13]
GLUT7 (*SLC2A7*)	Glucose and fructose	0.3 mM and 0.06 mM	small intestine, colon, testis, prostate	[12,13]
GLUT9 (*SLC2A9*)	Fructose and uric acid	urate ~0.6 mM	kidney, liver, small intestine, placenta, lungs, leukocytes	[13,17]
GLUT11 (*SLC2A11*)	Fructose and glucose	Fructose high affinity, and glucose low affinity	heart, skeletal muscle, kidney, adipose tissue, placenta, pancreas	[13]
CLASS III				
GLUT6 (*SLC2A6*)	Glucose	5 mM	brain, spleen, leukocytes, lysosomal membranes	[10,12,13]
GLUT8 (*SLC2A8*)	Glucose, fructose, DHA, and galactose	2 mM	testis, insulin-sensitive tissues	[13,14,16]
GLUT10 (*SLC2A10*)	Glucose and galactose	0.3 mM	liver, heart, pancreas, brain, ER glucose transporter	[8,16,18]
GLUT12 (*SLC2A12*)	Glucose, galactose, and fructose	Glucose high affinity	skeletal muscle, central nervous system, adipose tissue, small intestine, heart	[8,12,16]
HMIT (GLUT13) (*SLC2A13*)	Mio-Inositol H+- coupled	100 μM	brain	[13]

DHA: dehydroascorbic acid; ND: not described.

**Table 3 ijms-24-06085-t003:** Different GLUTs (SCLA2) involved in various types of cancers.

Tissue	Transporter Modified in Malignancy	Ref.
**Nervous system**		
Astrocytes	1, 3, 4, 12	[66,67,68]
Brain	1, 3	[66,68]
Brain–blood barrier	1	[69]
Glia	1, 3, 4, 5	[66,67,68,70]
Meninges	1	[70]
Oligodendrocytes	12	[66,69]
Retina	1	[70]
**Digestive system**		
Colon	1, 2, 4, 5, 12	[66,68,70]
Esophagus	1	[70]
Larynx	1, 3	[66,70]
Liver	1, 2, 5, 9	[68,70]
Mouth	1, 3	[68,70]
Pancreas	1, 2, 4, 5, 6	[66,67,68,70]
Rectum	1, 2, 5	[68,70]
Stomach	1, 2, 3, 4, 6, 10	[66,67,68]
Tongue	1	[70]
**Reproductive system**		
Breast	1, 2, 3, 4, 5, 6, 12, HMIT	[66,67,68]
Cervix	1	[66,67,68]
Endometrium	4, 6, 8	[67,68]
Ovary	1, 3,	[66,67,68]
Prostate	1, 3, 5, 7, 9, 11, 12	[66,67,68]
Testis	5, 6, 9, 14	[66,67,68,70]
Uterus	1, 6	[66,70]
**Respiratory system**		
Lung	1, 3, 5, 9, 12	[66,70]
Nasopharynx	1,	[68,70]
**Urinary system**		
Bladder	1, 3	[66]
Kidney	1, 5, 9, 13, HMIT	[66,68,70]
Urethra	1	[70]
**Immune system**		
Plasmatic cells	4, 7, 8, 10, 11	[66,67]
Lymphoid tissue	1, 4, 5, 8	[66,70]
**Dermal system**		
Melanocytes	1, 3	[68,70]
Skin	1, 3, 9	[66,68,70]
**Glandular system**		
Thyroid	1, 2, 3, 4, 9, 14	[66,70,71]
**Muscular system**		
Heart	1, 9	[68]
Muscle	1, 12	[68]
**Skeletal system**		
Bone	1, 3	[72]

**Table 4 ijms-24-06085-t004:** SLC50 family.

Transporter	Transports	Function	Refs
Clade I SWEETs	Hexoses		
SWEET1 (RAG1AP1) (*SLC50A1*)	Glucose	β-cells, pathogens, and symbionts	[87,88]
SWEET2 (*SLC50A2*)	2-Deoxyglucose	Plants	
SWEET3 (*SLC50A3*)	2-Deoxyglucose		
Clade II SWEETs	Hexoses		
SWEET4 (*SLC50A4*)	Glucose		
SWEET5 (*SLC50A5*)	Glucose		
SWEET6 (*SLC50A6*)	Glucose		
SWEET7 (*SLC50A7*)	Glucose		
SWEET8 (*SLC50A8*)	Glucose		
Clade III SWEETs	Sucrose		
SWEET9 (*SLC50A9*)	Sucrose		
SWEET10 (*SLC50A10*)	Sucrose		
SWEET11 (*SLC50A11*)	Sucrose		
SWEET12 (*SLC50A12*)	Sucrose		
SWEET13 (*SLC50A13*)	Sucrose		
SWEET14 (*SLC50A14*)	Sucrose		
SWEET15 (*SLC50A15*)	Sucrose		
Clade IV SWEETs	Fructose		
SWEET16 (*SLC50A16*)	Glucose, sucrose, fructose		
SWEET17 (*SLC50A17*)	Fructose		
SemiSWEETS			
BjSemiSWEET	Sucrose	Bacteria	
LbSemiSWEET	Glucose		
VsSemiSWEET	ND		
TySemiSWEET	ND		
EcSemiSWEET	Sucrose		

ND: not described; Adapted from [87].

**Table 5 ijms-24-06085-t005:** Plant and fungi sugar transporters.

Transporter	Subtypes	Sugar Type	Transports	Km	Ref.
SUT (Fungi)	1–5	Disaccharide	Sucrose and maltose H^+^/sucrose symporter	0.3–1 mM	[97,98]
SUC	1, 2, 3, 4, 5, 8,13				
STP	1–4, 6–14	Monosaccharide	Hexoses and pentose	10–100 µM	
MST	1, 5, 6				
HEX (Fungi)	1, 2, 3, 6				
ST	1				

## Data Availability

Not applicable.

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
