# Peer review of "Relevance of Sugar Transport across the Cell Membrane"

_ijms, 2023, doi:10.3390/ijms24076085_

Round 1

Reviewer 1 Report

Minor comments

Extensive English editing is a must.

Line33; " accumulate and feed", remove accumulate.

Line 34-36; "In mammals, sweetness perception is initiated when sweeteners interact with taste receptor proteins in taste buds, and sweet taste has a complex genetic architecture". Crack it to two sentences, the last part " and sweet taste has a complex genetic architecture" is not in lone with the first sentence.

Line 43; what is the meaning of " the cells must relate"? kindly change to appropriate sentence.

Line 61; Do NOT start a sentence with "And".

Line 72; What is the meaning of" and hormones". Is not Insulin an hormone?

Line 125; expand on each GLUT class. What are the main characters of each class?

Line 246-249; who is this paragraph related to the previous ones?

The subtitles: Diabetes, liver health, Kidney, Hypertension, Aging, bone health, sleep… are NOT consistent. Either use organs or health condition.

Line 452, " possible therapeutic targets", for what or which condition/deseas?

Line 455; " sugar transporters transport systems"?

Line 502-503; rewrite.

Line 585; " There is a relatively recent discovery". 2016 is Not a recent. Kindly revise and look for other recent reference (2020 or newer).

Major comments

Several Know researches of GLUTs and Diabetes are NOT mentioned or cited in this study like Klip A., James D.E. and McGraw T.E. Kindly refer to their published papers, add their main findings and cite them.

The subtitles: Diabetes, liver health, Kidney, Hypertension, Aging, bone health, sleep… are NOT consistent. Either use organs or health condition.

In several spots, the paragraphs are not well connected. Some examples are mentioned above. Yet, the authors are aught to revise the whole manuscript.

Author Response

Please see the attachment, Answer to reviewer 1

Reviewer 2 Report

The review submitted by Roxana Carbó and Emma Rodríguez summarizes the type of sugar transporters and their physiological and pathological relevance. Although there are already several reviews on the same topic, the authors present a more exhaustive revision of all sugar transporters, not focusing on only one type. However, the review in some aspects is not deep enough, and there are some concerns to solve before publication.

 1-     The organization can be improved. Presenting all the transporters and, then, their role in physiology and disease would facilitate the reading.

2-     I suggest improving the introduction, including the function of sugars in the body (energy production, glycosylation, etc.).

3-     Table 2. I missed the transport of dehydroascorbic acid for some of the GLUT transporters. I would add this information to the table and mention it in the text.

4-     In the Obesity and Diabetes section, you do not mention the importance of insulin-dependent transporters. Please, discuss and add the corresponding information.

5-     Line 319: Include the reference and the endpoint.

6-     The Immune system section needs to be further explored: I recommend including all the blood cell lineages (lymphocytes, macrophages, erythrocytes, hematopoietic Stem cells, etc.). I also suggest extending the importance of GLUT transporters in lymphocyte activation.

7-     The section devoted to cancer is scarce. I understand that the role of glucose transporters in cancer is not the major goal of the review, but the increase in GLUT expression is one of the hallmarks of cancer metabolism. I would introduce the metabolic reprogramming of cancer cells and the importance of the different transporters. To facilitate the reading, I consider including a table or scheme with the cancers in which each transporter is important. In addition, not only the factors including are regulators of GLUT expression in cancer. Here, I would also include a figure or table.

8-     I do not include a section only for Plants; I would add it with the SWEET transporters.

9-     I missed a section about targeting sugar transporters in clinics. Please, add it at the end of the manuscript.  

Author Response

Please see the attachment, Answer to reviewer 2

Round 2

Reviewer 1 Report

fine

Reviewer 2 Report

I thank the authors for addressing all the comments.